# Consumer Attitudes toward Consumption of Meat Products Containing Offal and Offal Extracts

**DOI:** 10.3390/foods10071454

**Published:** 2021-06-23

**Authors:** Mar Llauger, Anna Claret, Ricard Bou, Laura López-Mas, Luis Guerrero

**Affiliations:** 1Food Safety and Functionality Program, Institute of Agrifood Research and Technology (IRTA), Finca Camps i Armet s/n, 17121 Monells, Spain; mar.llauger@irta.cat (M.L.); ricard.bou@irta.cat (R.B.); 2Food Quality and Technology Program, Institute of Agrifood Research and Technology (IRTA), Finca Camps i Armet s/n, 17121 Monells, Spain; anna.claret@irta.cat (A.C.); laura.lopezm@irta.cat (L.L.-M.)

**Keywords:** theory of planned behavior, viscera, valorization, by-products, sustainability, consumer perception, consumer attitude

## Abstract

The development of food products containing offal and offal extracts could be part of the solution to the upcoming demand for animal protein. This study aimed to determine Spanish consumers’ attitudes toward offal and the development of meat products containing offal extracts. Consumers’ perceptions were evaluated by means of focus group discussions and a survey (*N* = 400) to validate the focus group results in various Spanish provinces. The theory of planned behavior was used to examine consumer attitudes. Results indicated that nutritional properties, environmental sustainability, and affordability were the main drivers, while sensory attributes, low frequency consumption, and perceived higher content of undesirable compounds were the main barriers. Three segments were identified according to their beliefs: those in favor of these products, those that were health and environmentally conscious, and those who were reluctant about them. The identification of these segments and their profiles demonstrated the necessity to focus efforts on providing reliable information on sensory and health-related issues to improve acceptability. Attitude was the most important predictor of behavioral intention regarding the global model, while the social component (subjective norm) was significant for two of the identified segments, emphasizing the relevance of the social component for acceptability.

## 1. Introduction

It is expected that in the next decade, meat consumption levels in developed countries will remain high, whereas in the developing countries of Asia and Latin America, meat demand is expected to increase fourfold [1]. Growing global meat consumption is a result of food system globalization [2], demographic changes [3], and, in some developing countries, nutritional needs to consume foods with a higher content of animal protein [4]. Consumers’ preferences and sensory properties also play an important role in the growing demand of meat products [5]. 

The global demand for animal protein sources has a negative impact on the environment [6]. Slaughterhouses also generate large amounts of waste and animal by-products [7,8]. A distinction can be made within animal by-products: those that are inedible (e.g., hair, horns, teeth, glands) and those that are edible, such as various organs (e.g., gizzards, heart, kidneys, liver) and commonly referred to as offal. In a world with finite resources, the minimization, recovery, and utilization of edible animal by-products may not only serve to decrease the environmental impact, but also significantly reduce the processing costs within the meat industry supply chain. In general, edible animal by-products are a valuable resource with high nutritional value, due to their high protein and low fat levels, as well as good vitamin and mineral content [9,10]. However, it is worth mentioning that meat from edible organs can be considered a food source, depending on household budget, country regulations, and cultural heritage. For instance, in certain European countries and the southern parts of the United States, chitterlings, trotters, thymus, testicles, tongue, snout, and other offal meat from livestock are common menu items. 

In general, the use of edible animal by-products for human consumption in Europe has decreased throughout the 21st century [11]. In Spain, offal consumption per capita decreased from 1.15 kg to 0.86 kg over the period 2004–2014 [12]. This trend can be linked to various factors, including dietary changes, the increasing demand for convenient products [13], risk of health hazards (i.e., bovine spongiform encephalopathy) [14], heavy metal accumulation (i.e., Cd and Pb) [15], and their association with low-income consumers [16]. Therefore, understanding consumers’ perceptions is key to identifying the main drivers and barriers behind offal consumption. It would then be possible to valorize offal by transforming it into more convenient foods that are adapted to the current food consumption lifestyles, or by developing new functional ingredients for the food industry [17,18]. Hence, different protein extracts from offal have already been applied to the development of Frankfurters [19,20] representing a sustainable strategy for circular bioeconomy. 

Food choice can be influenced by multiple factors, such as context (e.g., physical and social surroundings), biological (innate), cultural determinants, individual/psychological experience [21], and quality perception. One of the most widely used models to examine the relationship between beliefs, attitudes, and behavior to predict consumer behavioral intention is the theory of planned behavior (TPB) [22,23,24]. This theory is an extension of the theory of reasoned action [25] combined with perceived behavioral control, as a measure of individual intention for performing a behavior. In both approaches, behavior is predicted by behavioral intention, which can be assessed through the TPB model. Behavioral intention is predicted by the personal attitude toward the behavior, the subjective norm (i.e., people’s beliefs about what other important people think they should do), and the addition of perceived behavioral control as a measure of perception of the ease or difficulty toward performing the behavior of interest. 

The goal of this study is to explore the main drivers and barriers behind offal consumption and measure consumer attitudes toward the development of meat products containing offal extracts by means of the TPB.

## 2. Materials and Methods

The study is organized into two sequential stages: (1) a qualitative exploratory approach, by means of focus groups aimed at assessing consumer perceptions and beliefs on the consumption of offal and its possibility of use as ingredients in the development of meat products; and (2) a quantitative approach by means of a survey, to validate the results previously obtained through the focus groups by using the TPB model.

### 2.1. Qualitative Stage: Belief Selection

Five focus group discussions were conducted in four Spanish geographic locations: one focus group each in Madrid, Seville, and Barcelona, and two focus groups in Girona. In each focus group, eight participants were selected, as per their gender (50% men and 50% women), age (between 20 and 65 years), and eating habits (at least three participants from each focus group consumed offal, two or more times per month). The focus group sessions were structured into three stages. The first stage consisted of a general discussion of meat and edible animal by-products as fresh products (e.g., liver, kidneys) or as ingredients (e.g., pate). The second stage consisted of a general discussion on the advantages and disadvantages of meat and offal consumption. Finally, the third stage involved a general discussion about the participants’ perceptions of the use of edible by-products as ingredients or extracts in the development of meat products. In addition, the perception of health-related issues regarding the consumption of edible by-products was considered. Each focus group session was conducted by the same expert moderator, following the recommendations of Guerrero and Xicola [26] and Krueges and Casey [27]. Sessions lasted about 90 minutes and were audio and video recorded for a deeper analysis. The most relevant beliefs about using edible by-products as ingredients or extracts in meat products were identified and retained in the design of the quantitative questionnaire. 

### 2.2. Quantitative Stage: The Theory of Planned Behavior

To measure consumer attitudes toward the development of meat products containing offal extracts, a questionnaire was developed according to Ajzen’stheory of planned behavior [22]. It consisted of 32 questions and included nine items on behavioral beliefs extracted from the qualitative stage (i.e., the focus groups) and their corresponding evaluations (very bad/very good), three items on attitude, three items on normative beliefs and their corresponding items on motivation to comply, one item on subjective norm, two items on perceived control, and two items on behavioral intention. Table 1 shows the structure of the questionnaire, including the items assessed and the scoring scale used for each of them. All questions were randomly mixed into the final questionnaire for a less biased assessment of internal consistency, which was measured using Cronbach’s alpha coefficient [28]. Some items of the TPB model were reversed in the questionnaire, to avoid the “yea-saying” and “nay-saying” response bias [29], and then transformed again in the right direction for data analysis.

Four hundred individuals were recruited from various Spanish provinces (Table 2) per quota (gender and age) by using convenience sampling. The education level, perceived economic situation, and information on consumption frequency of offal (liver and kidneys) were also obtained during the recruitment of the participants. 

### 2.3. Data Analysis

To analyze the data obtained, all the items of the TPB model that were assessed by using a 7-point Likert scale were transformed from −3 to +3, with +3 representing a positive view. The only exceptions were the motivational items, which were scored from 1 to 7, as suggested by Ajzen and Fishbein [30]. In addition, some originally negative statements of behavioral beliefs (numbers 4, 6, 7, and 8 of Table 1) in the questionnaire provided to participants were transformed into positive beliefs to facilitate the understanding of the results. 

Cronbach’s alpha coefficient was used. According to Cronbach [28], this provides for internal consistency; thus it was used for the different multi-item compounds of the TPB model. The unitary structure of all the multi-item compounds of the TPB model were assessed through factor analysis that used the principal component method [31]. The relationship between the sum of behavioral beliefs multiplied by their evaluations, the sum of normative beliefs multiplied by their motivation to comply, the sum of attitude items, and the subjective norm were all assessed through Pearson’s correlation coefficient. Multiple linear regression was applied to some of the different components of the TPB model (attitude, subjective norm, and perceived control) to determine their ability to predict behavioral intention. In all cases, the absence of multicollinearity was checked beforehand [32]. 

Series of one-way analysis of variance (ANOVA) were carried out to examine the existence of differences for each component of the TPB model and for each belief (both behavioral and normative), depending on the gender, age, education level, income, and consumption frequency of offal (liver and kidneys). Tukey’s honestly significant difference post hoc test was applied to assess the statistical differences among the selected dependent variable levels. 

Finally, an agglomerative hierarchical cluster analysis using Ward’s method and Euclidian distance was performed on the behavioral beliefs, to identify segments of consumers with similar belief patterns. According to Hair et al. [32], the number of segments to be retained was selected based on the obtained dendrogram, by considering the homogeneity within and among the segments and the principle of parsimony [33]. Discriminant analysis was performed to validate the number of clusters retained, by checking the number of individuals who were properly classified into their corresponding cluster (i.e., the confusion matrix). A one-way ANOVA was carried out to determine the significant differences between the behavioral and normative beliefs and the different components of the model. Finally, to characterize the various clusters obtained, an additional one-way ANOVA (with cluster as the dependent variable) was performed for the quantitative sociodemographic variables, with a chi-squared test for the qualitative variables (i.e., gender and consumption frequencies). 

Data were analyzed using the XLSTAT statistical software, version 19.6 (2020) (Addinsoft, France).

## 3. Results and Discussion

### 3.1. Characteristics of the Sample

The final sample included 200 men and 200 women, between 20 and 63 years of age. The age range distribution was: 8.3% of participants were 16–24 years, 27.0% were 25–34, 33.3% were 35–44, 25.0% were 45–54, and 6.5% were 55–64. Both gender and age distributions matched the national average [34]. The education level showed a bias toward highly educated consumers (57.3%) when compared to the national average (30.6%), in detriment of the medium education level, which was lower (33.8%) than the national average (50.5%) [35]. This bias was probably due to the higher self-confidence levels and the willingness to take part when people had a higher education level, as reported by Claret et al. [36].

### 3.2. Qualitative Approach: Focus Groups

The focus groups discussions provided relevant insights into the salient beliefs about meat products containing offal extracts and how these perceptions might influence their food choices. Most participants (75.0%) had positive beliefs overall about the nutritional properties of offal, especially regarding their high content in iron and vitamins. For several years, the mass media in Spain have been emphasizing the iron supply associated with the consumption of liver and pate, which could explain the observed prevalence of this belief.

Although the consumption of offal has decreased among the Spanish population, it is still present in the Spanish diet. Viscera and offal represented 0.51% of the iron dietary source among the Spanish population [37].

However, participants expressed certain barriers to offal consumption that were related to their appearance and odor. Offal products were described as “unsightly”, “viscous”, and having a “fluffy texture” and “unpleasant odor”, especially when they were raw products. The negative visual and odor sensory attributes influenced the acceptance or rejection of food products, and may influence acceptance much more than taste [38,39]. This insight could explain the fact that the overall perception improved when offal was consumed as an ingredient in other types of meat products, such as pate. In this sense, a favorable perception of spreadable liver paste could be linked to widespread pate consumption in Spain (0.35 kg per capita in 2006) [40]. Therefore, past life experiences or the frequency of usage could increase the acceptability of meat products containing offal extracts. According to a Dutch study [41], habitual consumption was a strong determinant of food choice, as respondents stated that “they were eating the way that they were taught at their parents’ home and continued eating according to those habits when they were to live on their own”. Those participants with a low frequency consumption of offal perceived as a barrier for consumption the required cleaning step, either due to ignorance or the time it would take. Consumers’ food choices are linked to the product’s convenience, as reported by Scholderer and Grunert [42].

In general, consumers’ safety beliefs about offal were negative, mainly because they perceived a higher content of undesirable compounds (e.g., toxins or drug residues), when compared to other meat products. Food safety, being interpreted as the need to guarantee the non-toxicity of foodstuffs, is a food quality that consumers expect when purchasing a food product, and affects the consumer decision-making process [43,44]. Hormone or veterinary drug residues, chemical environmental contaminants, or microbial pathogens increase consumers’ risk perceptions and decrease consumer confidence [45]. In this sense, participants’ perceptions of safety were an important barrier in the reported behavioral intention toward offal consumption observed in the focus group discussions. 

The growing concern about environmental protection has increased the importance of food purchase sustainability [46]. Therefore, environmental aspects were included in the focus group discussions. Awareness of the environmental impact of meat production systems and the influence of by-product usage in food waste reduction were discussed. Participants held positive attitudes toward the development of new strategies that improved the reduction and valorization of offal (e.g., meat products containing offal extracts). This attitude is supported by the Mintel Global Food and Drink Trends report, which points out consumers’ concerns about the environmental and ethical impacts of their diets, mainly because they are looking for friendly production practices and sustainable diets, a trend that is expected to continue in the next decade [47]. Broadly speaking, similar insights were obtained in the five focus group discussions conducted in various Spanish locations. 

These results correspond to those reported by Henchion et al. [48], who used focus groups to investigate consumer evaluations of food products that incorporated ingredients derived from beef by-products. These authors reported that physical state, perceived naturalness of the ingredients, and past life experiences are factors that significantly influence their acceptance. In this study, the focus group results were used to develop a questionnaire that assesses consumers’ attitudes toward using offal extracts for the development of meat products.

### 3.3. Quantitative Approach: Survey

The Cronbach’s alpha coefficient for multi-item compounds of the TPB model (behavioral beliefs × evaluation, normative beliefs × motivation to comply, attitude, perceived control, and behavioral intention) ranged from 0.92 for the perceived control measure to 0.71 for behavioral beliefs × evaluation items (Figure 1). Overall, these values showed good internal reliability [49]. Additionally, the factorial analysis verified the unitary structure of all the multi-item compounds of the TPB model, except for behavioral and normative beliefs. This fact indicates that not all belief items contribute in the same direction to explain their corresponding constructs [50,51]. Although beliefs tend to be internally consistent with one another [52], the non-unitary structure of the two model components suggests that people may hold beliefs that are not completely consistent. However, these inconsistencies do not represent a major problem, as stated by Sheperd [53] and Guàrdia et al. [54]. Therefore, the analysis of the beliefs is presented in both individual and aggregated manner in Table 2 and Table 3. 

The ANOVA of the six TPB components and the items of the two non-unitary components for each sociodemographic characteristic and offal consumption category reveal different consumer insights (Table 2). 

#### 3.3.1. Spanish Consumer Beliefs

With respect to behavioral beliefs, significant sociodemographic differences were found for all variables except education level (Table 2). In general terms, and according to the overall mean of behavioral beliefs, respondents showed negative opinions about the consumption of meat products containing offal extracts in the categories of sensory attributes (smell, taste, texture, and pleasantness), health issues, natural appearance, and safety issues (including toxins, drug residues, antibiotics, etc.). However, their beliefs were significantly more positive in relation to nutritional values, environmental issues (i.e., reducing food waste and environmental impacts), and, especially, affordability. 

These overall beliefs corresponded to the focus group results, highlighting the importance of sensory properties in the development of products containing offal. This could correlate to the fact that food is seen as a source of enjoyment in societies without food shortages [55]. However, even if consumer preferences are predominantly dependent on the sensory attributes of foods [56], people’s food preferences are more complex and dependent on other aspects, such as culture, attitudes, values, beliefs, and cooking practices [57,58].

The findings regarding gender showed significant differences in three of the nine items. In general, men had a more positive view of sensory attributes, a fact that could be explained by the differences in food preferences between men and women. It has been reported that men have a greater preference for foods with strong and rich tastes, high color intensities, and chewier textures, while women prefer pale and light foods with no troublesome textural properties [59,60]. The presence of blood and raw meat was negatively associated with living animals and, therefore, caused the dislike for meat consumption, especially among women. In addition, some Spanish regions have a widespread habit of “tapas” consumption, an appetizer that is consumed in bars and restaurants, which is sometimes prepared with offal (e.g., tripe). This practice is more common in men than in women. 

Men also held the strongest beliefs regarding the potential impact on food waste reduction when consuming meat products with offal extracts. However, the existing literature indicates that women are usually more concerned about environmental issues than men, based on gender roles and socialization [61,62]. 

Regarding age, the only significant differences were related to waste consciousness. Respondents between 35 and 44 years were more waste-conscious than those between 55 and 64 years. The effect of age on environmental concerns is affected by sociopolitical and socioeconomic variables. In general, age is negatively correlated to environmental concerns [63,64,65], as observed in the present study.

Income levels showed a significant difference in only three of the nine items. Respondents with a low income had a slightly negative view of nutritional value and were less eco-conscious compared to the well-off respondents. Environmental concerns are usually associated with socioeconomic status. In this sense, those who were economically disadvantaged tended to prioritize economic goals, in detriment of environmental protection [66,67]. 

According to the consumers’ self-reported behavior, there were significant differences in the frequency of offal consumption. Regular offal consumers (e.g., liver, kidneys) had a more positive view of sensory attributes, nutritional value, sustainability, and affordability. This perception can partly be explained through a higher degree of product knowledge regarding intrinsic (e.g., flavor, appearance, etc.) and extrinsic (e.g., place of origin, context of consumption, etc.) attributes that affect consumer beliefs [68,69]. Additionally, the association with familiar products can enhance the acceptance of unfamiliar foods. Research on the acceptability of new foods, such as insect-based foods and cultured meats, highlighted the role of using familiar ingredients or products to increase food acceptability and the willingness to eat it [70,71]. 

#### 3.3.2. Spanish ConsumerNormative Beliefs

The overall mean values in Table 2 regarding normative beliefs suggest that respondents believe that none of the relevant social groups that were explored (family, health personnel, and friends) would recommend that they consume meat products containing offal extracts. Eating is usually a social act that may affect the type of foods consumed. The relationships between individuals and their family, friends, and other people are important contexts influencing food choices within their social network [72]. Further, mean values of the motivation to comply considered to determine the extent by which these groups hold influence. The mean values indicate that family, health personnel, and friends have a certain influence on consumers’ food-related behaviors (4.57, 5.32, and 4.67 in a 7-point Likert scale; data not shown). In this sense, the influence of explored relevant groups, especially health personnel’s advice, could play an important role in the acceptance of offal-based products. Some studies suggest that health-related information or environmental benefits could increase the acceptability of insect-based foods and change consumer attitudes toward unfamiliar foods [73,74]. Therefore, providing health and environmental information through significant others could be a good strategy to increase acceptance. 

All sociodemographic variables besides gender had a significant effect on normative beliefs. The respondents who consumed liver and kidneys more often had stronger positive beliefs about other people’s recommendations. These beliefs may explain why these respondents continued to consume the liver and kidneys. Thus, they might assume that their direct social environment would not reject their offal eating habits. Age, education, and income also affected the strength of normative beliefs. The existing differences between categories, although significant, were not as relevant.

#### 3.3.3. Components of the TPB Model

Regarding the six components of the TPB model, there were significant differences in all sociodemographic variables except for the age categories (Table 2). Regarding the overall mean, behavioral beliefs × evaluation showed a positive mean value, whereas normative beliefs × motivation to comply were clearly negative. The remaining components (attitude, subjective norm, perceived control and behavioral intention) also showed negative values. These results indicate that even when respondents did not have strong negative behavioral beliefs, they expressed a negative overall intention, probably because the strength of the observed barriers (i.e., mainly sensory properties, safety, and health-related issues) had a greater impact than the positive outcome of performing the behavior (i.e., nutritional benefits and environmental issues). 

Women had a more negative attitude (−1.77 vs. −0.63) and lower behavioral intentions (−1.59 vs. −0.73) toward the consumption of meat products containing offal extracts than men. Women also showed a lower perceived control of the behavior (−1.63 vs. −0.86). This result is difficult to explain because women do most of the food shopping, as stated by Claret et al. [50]. The difference in perception between both genders may be due to the greater knowledge that women may possess about the actual availability of offal and derivates on the market.

Results showed that respondents with a high education level were less willing to do what significant others thought they should do, according to subjective norm (−0.99 vs. −0.58 and 0.03). This was similar to the observation of the normative beliefs × motivation to comply (−13.27 vs. −8.84 and 2.81) data. These results seem to indicate that people with higher education levels tended to have higher established beliefs and are less affected by external opinions and recommendations than those with lower education levels [54]. 

Respondents with a higher income had a more positive value for behavioral beliefs × evaluation (13.57 vs. 3.84 and 1.96) and were more greatly affected by significant others, considering the mean values for the normative beliefs × motivation to comply (6.30 vs. −11.21 and −13.00) and subjective norm (0.17 vs. −0.81 and −0.93). They also had more positive attitude toward the product (1.52 vs. −1.26 and −2.64), higher perceived control (0.57 vs. −1.29 and −2.18), and higher behavioral intention (0.78, −1.25, and −1.64). In any case, it is important to note that even when significant, the number of individuals in the well-off and difficult groups (5.8% and 7.0% of the respondents, respectively) are rather small for drawing valid generalizations. Finally, according to the self-reported consumption of liver and kidneys, the results showed that respondents with regular offal consumption had higher positive mean values than those who consumed offal less than once a month, thus highlighting the importance of food habits or past experiences. It is worth mentioning that the number of participants who consumed offal regularly is rather low compared to those who consume it only occasionally (28.3% and 13.5% for liver and kidneys, respectively).

#### 3.3.4. Cluster Analysis

The cluster analysis results of the six TPB components and the items of the two non-unitary elements are shown in Table 3. Three clusters were retained, according to the discriminant analysis that was performed (93.0% of the participants were correctly classified in their respective clusters, according to the confusion matrix). Those clusters were labeled according to the participants beliefs as: “pro-offal-based meat products”, “health and environmentally consciousness”, and “reluctant to consume offal-based meat products”. The first cluster represented 50.5% of respondents, the second cluster represented 23.5%, and the third represented 26.0%. All clusters were significantly different for each item (behavioral and normative beliefs) or components of the assessed model. 

In reference to behavioral beliefs, respondents in the first cluster reported positive beliefs for all items. Thus, we referred to this first cluster as “pro-offal-based meat products”. The second cluster of respondents reported negative beliefs for health, safety issues (e.g., higher toxins, residue, and antibiotic content), and product appearance (e.g., less pleasant and less natural). Despite this fact, this cluster also reported the most positive beliefs for reducing food waste and environmental impacts. Thus, we referred to this second cluster as “health and environmentally consciousness”. Lastly, the third cluster of respondents reported overall negative views in eight of the nine assessed behavioral beliefs, especially regarding sensory attributes, nutritional value, and environmental issues. Therefore, we referred to this third cluster as “reluctant to consume offal-based products”. Finally, it is worth mentioning that participants from all clusters perceived meat products containing offal extracts as economically affordable, so it seems that consumers would expect a lower price for these types of meat products than for regular, similar products. According to de Magistris and Gracia [75], consumers are willing to pay a premium price for food products that are perceived as sustainable and beneficial for the local economy. This was not observed within the current study, probably because consumers attached more importance to the fact that offal was used in the product, rather than the fact that the resulting product was more sustainable.

Overall negative values were observed for normative beliefs. This trend was stronger in the third cluster, followed by the first and second clusters (–2.14 vs. −1.84 and −1.95, respectively). Surprisingly, the findings showed that, although the first cluster had overall positive beliefs for all the assessed items, they did not believe that their significant others would recommend that they consume these types of meat products. Indeed, it seems that this social barrier might be a problem for promoting the consumption of offal-based meat products. 

According to behavioral beliefs × evaluation, the first cluster had the strongest positive beliefs (8.10 vs. 6.35 and −5.05). This cluster also had the most positive attitude (0.64 vs. 0.00 and −5.84) and positive perceived control (0.16 vs. −0.65 and −4.51). However, the behavioral intention was slightly negative, suggesting that the purchase choice could be influenced more by significant others rather than their own beliefs or control over the behavioral control. The second cluster was characterized as being more neutral regarding attitude, subjective norm, and behavioral intention (0.00. −0.09, and 0.10, respectively).

However, just as with the first cluster, this cluster held positive beliefs regarding behavioral beliefs × evaluation. The second cluster was the only one where the Behavioral intention was slightly positive, in agreement with participants’ behavioral and normative beliefs. Finally, the third cluster reported the most negative values in all components of the TPB model. 

The characterization of clusters with respect to some sociodemographic variables showed significant differences regarding gender, income, and offal consumption frequencies (results not shown). The second cluster was characterized by intermediate-income respondents (*p* ≤ 0.05). The third cluster had a higher percentage of women (62.5%; *p* ≤ 0.01) and contained those with the lowest offal consumption frequency (with 90.4% and 95.2% consuming the product less than once a month, for liver and kidneys, respectively). Indeed, these findings might explain the worse perception of the respondents from the third cluster, because women showed more negative beliefs, mainly regarding sensory attributes.

#### 3.3.5. TPB Model

The final model obtained (Figure 1) for the pooled sample of participants showed good predictive capacity (R^2^ = 0.76) of behavioral intention toward the consumption of meat products containing offal extracts, highlighting the utility of the TPB model.

Behavioral intention was significantly correlated (*p* ≤ 0.001) with attitude, perceived control, and subjective norm (r = 0.83, 0.76, and 0.76, respectively). These correlational values were consistent with the good predictive capacity of the model. Normative beliefs × motivation to comply was significantly correlated to subjective norm (r = 0.84; *p* ≤ 0.001), and behavioral beliefs × evaluation was correlated to attitude, although this was to a lower extent (r = 0.61; *p* ≤ 0.001). This lowest correlation could be explained because having a positive attitude toward something does not imply agreeing with all beliefs, as they cover different dimensions (e.g., health issues, sensory attributes, environmental issues, etc.). 

Attitude was the most important element in predicting behavioral intention (standardized regression coefficient β = 0.45), followed by subjective norm, whereas perceived control was the least important (β = 0.15). Other studies have also reported attitude as a relevant predictor of intention for the consumption of novel foods and ready-to-eat meals (e.g., Mahon et al. [76]; Menozzi et al. [77]. The impact of perceived social effect (Subjective norm) on intention should be noted, a phenomenon already reported in other studies related to the purchase of sustainable and organic food [46]. In addition, the effect of perceived control on behavioral intention, although less important than attitude or subjective norm, also suggests that the ability to purchase meat products containing offal extracts affects the potential success of these products in the market. 

The individual model for each of the three identified clusters is also shown in Figure 1. Although the three individual models (one for each cluster) show good predictive capacity (R^2^_C_1__ = 0.68, R^2^_C_2__ = 0.88; R^2^_C_3__ = 0.61), the model for the second cluster was more efficient in predicting the behavioral intention, according to the TPB model. Regarding the standardized regression coefficient, the most important element in predicting behavioral intention was the subjective norm for the first and second clusters (β = 0.37 and β = 0.59, respectively), while attitude was the most important element for the third cluster (β = 0.55). Moreover, in the third cluster, the correlation between behavioral beliefs × evaluation and attitude was not significant, probably as a result of the adjectives used to measure the attitude, since this cluster had the strongest beliefs regarding environmental issues, which may not have been adequately captured by the attitudinal measurements. Furthermore, a non-significant result for the standardized regression coefficient regarding perceived control in the second and third clusters was obtained.

## 4. Conclusions

Overall, many respondents who participated in this study showed a negative attitude toward meat products containing offal extracts, mainly because of sensory and health-related concerns. Sensory properties are a crucial element in food choices, thus explaining that certain consumers are not willing to sacrifice sensory pleasure, even for an improvement in the healthiness of the product. A lower perception of sensory properties, safety, and healthiness of these products perfectly explains the negative attitude toward them, as observed in this study. However, a relevant barrier to consumption that was observed in all these segments was the social component; that is, the perception of significant others. Therefore, it is necessary to focus on marketing strategies for these products to reduce these negative effects by providing clear and reliable information on them and highlighting the positive aspects linked to their use. In any case, there is a noteworthy part of the population that clearly appears to favor this type of product, and who appreciates its sensory and nutritional properties. 

Current trends increasingly focused on the reduction of food waste, the use of by-products and, therefore, the increase in the sustainability of the food industry, constitute a promising scenario for meat products containing offal extracts. It is important to feed the growing population and search for new alternative sources of protein with high biological values. These present an opportunity for innovation and food industry competitiveness, by means of bioeconomic strategies.

## Figures and Tables

**Figure 1 foods-10-01454-f001:**
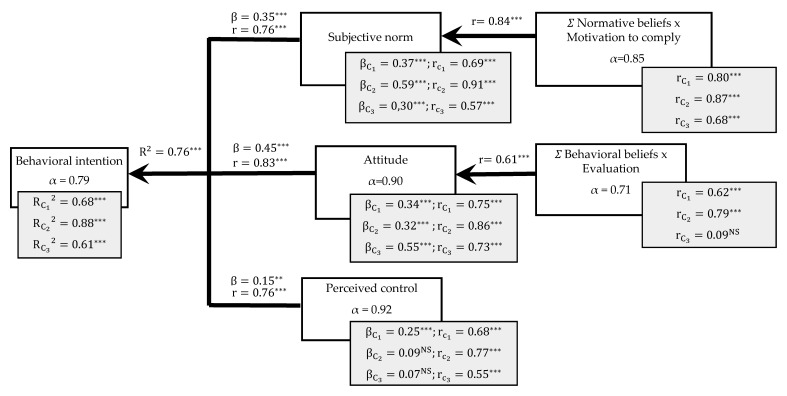
Final model of the theory of planned behavior for all the respondents (*N* = 400) and for each cluster of participants. Significance: *** *p* ≤ 0.001; ** *p* ≤ 0.01; r: regression coefficient; R^2^: coefficient of determination; β: standardized regression coefficient; α: Cronbach’s alpha coefficient. NS: Not Significant.

**Table 1 foods-10-01454-t001:** Structure of the questionnaire.

TPBComponents	Question Number	Question
*** Behavioral** **beliefs**		**In my opinion meat products containing offal extracts will …**
1	look good, taste good, smell good and have adequate texture
2	be more nutritious (more proteins, more iron, etc.)
3	be cheaper
4	be healthier (less cholesterol, less fatty, etc.)
5	help to reduce food waste
6	be pleasant
7	not contain more toxins, drug residues, antibiotics, etc.
8	be more natural
9	help reduce to some extent the environmental impact of animal production
**Normative** **beliefs**		**I think if I asked them …**
1	my family would recommend that I eat meat products containing offal extracts
2	my friends would recommend that I eat meat products containing offal extracts
3	my doctor would recommend that I eat meat products containing offal extracts
**Motivation to comply**		**In general, I try to follow what …**
1	my family may recommend
2	my doctor may recommend
3	my friends may recommend
**Subjective norm**	1	**In my opinion, most people who are important to me would recommend that I consume meat products containing offal extracts**
**Perceived** **control**		**If meat products containing offal extracts were available in the market …**
1	I think I could purchase them whenever I wanted
2	I would purchase and consume them whenever I felt like it
**Behavioral** **intention**		If meat products containing offal extracts were available in the market, …
1	there is a strong likelihood that I would consume them in the next few weeks
2	I would certainly consume them in the next few days
**Scale used:**	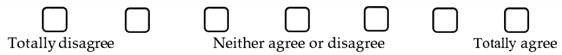
*** Evaluation** **beliefs**		**To me, that meat products containing offal extracts …**
1	look good, smell good and have adequate texture is …
2	are more nutritious (more proteins, more iron, etc.) is …
3	are cheaper is …
4	are healthier (less cholesterol, more fatty, etc.) is …
5	help to reduce food waste is …
6	are pleasant is …
7	do not contain more toxins, drug residues, antibiotics, etc. is …
8	are more natural is …
9	help to reduce into some extent the environmental impact of animal production is …
**Scale used:**	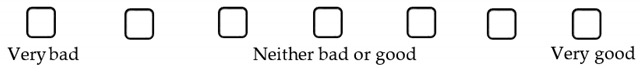
**Attitude**		**In general, my consumption of meat products containing offal extracts would be …**
1	Bad/Neither good nor bad/Good
2	Disgusting/Neither disgusting nor pleasant/Pleasant
3	Harmful/Neither harmful nor beneficial/Beneficial

* Behavioral beliefs and their corresponding evaluation belief, numbered 4, 6, 7, and 8, were formulated in a negative way for the original questionnaire and then transformed into positive statements to facilitate an under-standing of the results.

**Table 2 foods-10-01454-t002:** Least squared means for the components of the model according to sociodemographic characteristics and offal consumption of the respondents (*N* = 400).

															Offal Consumption							
		Gender		Age Category					Education			Income			Liver				Kidneys			
	Overall Mean	Men (n = 200)	Women (n = 200)	16–24 (n = 33)	25–35 (n = 108)	34–44 (n = 133)	45–54 (n = 100)	55–64 (n = 26)	High (n = 229)	Medium (n = 135)	Elementary (n = 36)	Well-Off (n = 23)	Intermediate (n = 349)	Difficult (n = 28)	One to Seven Times a Week (n = 24)	Once Every Two Weeks (n = 43)	Once a month (n = 46)	Less Than Once a Month (n = 287)	One to Seven Times a Week (n = 10)	Once Every Two Weeks (n = 19)	Once a Month (n = 25)	Less Than Once a Month (n = 346)
**Behavioral beliefs**																						
**1_Sensory attributes ^F^**	**−0.06 ****	**0.18 ^a^**	**−0.29 ^b^**	−0.21	−0.02	−0.04	−0.06	−0.08	−0.19	0.13	0.14	0.48	−0.06	−0.43 ***	**0.71 ^a^**	**0.93 ^a^**	**0.65 ^a^**	**−0.38 ^b^*****	**1.40 ^a^**	**0.84 ^a^**	**0.52 ^ab^**	**−0.19 ^b^**
**2_Nutritional ^F^**	**0.30**	0.34	0.26	0.06	0.48	0.25	0.30	0.08	0.26	0.28	0.58 *	**0.83 ^a^**	**0.31 ^ab^**	**−0.25 ^b^*****	**0.83 ^a^**	**1.19 ^a^**	**1.00 ^a^**	**0.01 ^b^*****	**1.50 ^a^**	**1.05 ^ab^**	**0.96 ^ab^**	**0.17 ^b^**
**3_Cheaper ^F^**	**1.04**	1.03	1.06	1.24	0.93	1.17	0.94	1.00	1.07	0.98	1.11	0.96	1.02	1.36 *	**1.58 ^a^**	**0.53 ^b^**	**1.09 ^ab^**	**1.06 ^ab^**	1.60	0.84	1.00	1.04
**4_Healthier *^,F^**	**−0.53**	−0.58	−0.48	−0.82	−0.48	−0.58	−0.42	−0.54	−0.49	−0.50	−0.86	−0.96	−0.52	−0.36	−0.63	−0.35	−0.35	−0.58	−1.40	−0.37	−0.76	−0.50
**5_Food waste reduction ^F^**	**0.23** *	**0.45 ^a^**	**0.02 ^b^***	**0.18 ^ab^**	**0.16 ^ab^**	**0.55 ^a^**	**0.09 ^ab^**	**−0.46 ^b^**	0.17	0.29	0.44	0.43	0.22	0.21 ***	**0.67 ^ab^**	**0.79 ^a^**	**0.93 ^ab^**	**0.00 ^b^**	1.10	1.00	0.52	0.14
**6_Pleasant to taste *^,F^**	**−0.64 ***	**−0.46 ^a^**	**−0.82 ^b^**	−0.97	−0.81	−0.49	−0.57	−0.54	−0.68	−0.53	−0.83	−1.30	−0.58	−0.79 **	**−0.63 ^ab^**	**0.12 ^a^**	**−0.37 ^ab^**	**−0.80 ^b^**	−0.80	−0.26	−0.08	−0.70
**7_Less toxins and other contaminants *^,F^**	**−0.48**	−0.44	−0.52	−0.73	−0.29	−0.56	−0.46	−0.65	−0.50	−0.40	−0.64	−1.17	−0.45	−0.29	−0.58	−0.12	−0.37	−0.54	−1.10	−0.79	−0.28	−0.46
**8_Natural *^,F^**	**−0.28**	−0.34	−0.23	−0.67	−0.27	−0.17	−0.38	−0.04	−0.18	−0.37	−0.61 *	**−1.13 ^b^**	**−0.20 ^a^**	**−0.61 ^ab^**	−0.75	0.07	−0.39	−0.28	−1.20	−0.89	−0.24	−0.23
**9_Reduction of the environmental impact ^F^**	**0.35**	0.40	0.30	0.70	0.43	0.4	0.21	−0.19	0.39	0.31	0.19 **	**1.00 ^a^**	**0.34 ^ab^**	**−0.14 ^b^*****	**1.00 ^a^**	**0.67 ^ab^**	**0.89 ^a^**	**0.15 ^b^*****	**1.50 ^a^**	**1.42 ^a^**	**0.44 ^ab^**	**0.25 ^b^**
**Normative beliefs**																						
**1_Family ^F^**	**−0.92**	−0.76	−1.08	−1.00	−1.04	−0.73	−0.89	−1.38 ***	**−1.10 ^b^**	**−0.87 ^b^**	**0.06 ^a^****	**0.04 ^a^**	**−0.93 ^b^**	**−1.57 ^b^*****	**0.04 ^a^**	**−0.12 ^a^**	**−0.15 ^a^**	**−1.24 ^b^*****	**0.70 ^a^**	**0.63 ^a^**	**0.12 ^a^**	**−1.12 ^b^**
**2_Friends ^F^**	**−0.84**	−0.74	−0.93 *	**−0.82 ^ab^**	**−0.74 ^a^**	**−0.64 ^a^**	**−0.96 ^ab^**	**−1.77 ^b^****	**−1.03 ^b^**	**−0.67 ^ab^**	**−0.17 ^a^****	**0.30 ^a^**	**−0.91 ^b^**	**−0.89 ^b^*****	**0.42 ^a^**	**−0.40 ^a^**	**−0.20 ^a^**	**−1.11 ^b^*****	**1.10 ^a^**	**0.53 ^a^**	**0.40 ^a^**	**−1.05 ^b^**
**3_Health personnel ^F^**	**−0.65**	−0.57	−0.73	−0.45	−0.68	−0.54	−0.75	−0.88 *	**−0.81 ^b^**	**−0.53 ^ab^**	**−0.03 ^a^***	**0.30 ^a^**	**−0.69 ^b^**	**−0.82 ^b^*****	**0.50 ^a^**	**0.00 ^a^**	**0.07 ^a^**	**−0.95 ^b^*****	**1.00 ^a^**	**0.58 ^a^**	**0.16 ^a^**	**−0.82 ^b^**
**TPB components**																						
**Behavioral beliefs × Evaluation ^A^**	**4.27**	4.83	3.71	−0.18	5.48	4.62	4.51	2.12	2.34	6.64	7.61 *	**13.57 ^a^**	**3.84 ^b^**	**1.96 ^b^*****	**12.58 ^a^**	**11.56 ^a^**	**12.33 ^a^**	**1.19 ^b^*****	**15.90 ^ab^**	**17.05 ^a^**	**7.76 ^a^**	**2.98 ^b^**
**Normative beliefs × Motivation to comply ^B^**	**−10.33**	−8.95	−11.71	−9.88	−10.62	−7.93	−10.61	−20.88 ***	**−13.27 ^b^**	**−8.84 ^b^**	**2.81 ^a^****	**6.30 ^a^**	**−11.21 ^b^**	**−13.00 ^b^*****	**6.29 ^a^**	**−1.40 ^a^**	**0.59 ^a^**	**−14.81 ^b^*****	**17.70 ^a^**	**12.16 ^a^**	**3.96 ^a^**	**−13.41 ^b^**
**Attitude ^C^**	**−1.20 ***	**−0.63 ^a^**	**−1.77 ^b^**	−2.03	−0.91	−0.98	−1.19	−2.46	−1.54	−0.89	−0.14 **	**1.52 ^a^**	**−1.26 ^b^**	**−2.64 ^b^*****	**1.83 ^a^**	**1.88 ^a^**	**1.26 ^a^**	**−2.30 ^b^*****	**3.60 ^a^**	**2.68 ^a^**	**1.84 ^a^**	**−1.77 ^b^**
**Subjective norm ^D^**	**−0.76**	−0.63	−0.89	−0.70	−0.79	−0.63	−0.83	−1.08 ***	**−0.99 ^b^**	**−0.58 ^ab^**	**0.03 ^a^***	**0.17 ^a^**	**−0.81 ^b^**	**−0.93 ^b^*****	**0.50 ^a^**	**−0.02 ^a^**	**0.00 ^a^**	**−1.09 ^b^*****	**0.80 ^a^**	**0.74 ^a^**	**0.28 ^a^**	**−0.96 ^b^**
**Perceived control ^E^**	**−1.25 ***	**−0.86 ^a^**	**−1.63 ^b^**	−1.27	−1.17	−1.20	−1.10	−2.31	−1.59	−0.75	−0.89 *	**0.57 ^a^**	**−1.29 ^b^**	**−2.18 ^b^*****	**0.46 ^a^**	**1.21 ^a^**	**0.67 ^a^**	**−2.06 ^b^*****	**1.40 ^a^**	**1.89 ^a^**	**0.64 ^a^**	**−1.63 ^b^**
**Behavioral intention ^E^**	**−1.16 ****	**−0.73 ^a^**	**−1.59 ^b^**	−1.52	−1.03	−0.79	−1.33	−2.54 **	**−1.59 ^b^**	**−0.76 ^b^**	**0.06 ^a^****	**0.78 ^a^**	**−1.25 ^b^**	**−1.64 ^b^*****	**1.08 ^a^**	**0.63 ^a^**	**0.57 ^a^**	**−1.90 ^b^*****	**2.40 ^a^**	**1.26 ^a^**	**1.28 ^a^**	**−1.58 ^b^**

Significance: *** *p* ≤ 0.001; ** *p* ≤ 0.01; * *p* ≤ 0.05. The 7-point Likert scale were converted from –3 to a +3, with the sole exception of motivational items. * Behavioral beliefs 4/6/7/8 were transformed to positive statements. ^A^ Behavioral beliefs × evaluation ranged between −81 and +81. ^B^ Normative beliefs × motivation to comply ranged between −63 and +63. ^C^ Attitude ranged between −9 and +9. ^D^ Subjective corm ranged between −3 and +3. ^E^ Perceived control and behavioral intention ranged between −6 and +6. ^F^ Behavioral beliefs and normative beliefs ranged between −3 and +3. The lower-case superscript letters indicate the significant differences for each of the sociodemographic characteristics.

**Table 3 foods-10-01454-t003:** Least squared means for the different components of the model by each of the three clusters obtained.

	Cluster 1	Cluster 2	Cluster 3
	“Pro-Offal-Based Meat Products” (n = 202)	“Health and Environmentally Consciousness” (n = 94)	“Reluctant to Consume Offal-Based Meat Products” (n = 104)
**Behavioral beliefs**			
**1_Sensory attributes ^F^**	0.79 ^a^	0.18 ^b^	−1.91 ^c^
**2_Nutritional ^F^**	0.80 ^a^	0.73 ^a^	−1.08 ^b^
**3_Cheaper ^F^**	0.70 ^b^	1.88 ^a^	0.93 ^b^
**4_Healthier *^,F^**	0.12 ^a^	−1.29 ^b^	−1.11 ^b^
**5_Food waste reduction ^F^**	0.61 ^b^	1.28 ^a^	−1.44 ^c^
**6_Pleasant to taste *^,F^**	0.34 ^a^	−1.83 ^b^	−1.46 ^b^
**7_Less toxins and other contaminants *^,F^**	0.04 ^a^	−1.57 ^c^	−0.51 ^b^
**8_Natural *^,F^**	0.02 ^a^	−0.93 ^b^	−0.30 ^a^
**9_Reduction of the environmental impact ^F^**	0.65 ^b^	1.24 ^a^	−1.06 ^c^
**Normative beliefs**			
**1_Family ^F^**	−0.63 ^b^	−0.18 ^a^	−2.14 ^c^
**2_Friends ^F^**	−0.68 ^b^	−0.05 ^a^	−1.84 ^c^
**3_Health personnel ^F^**	−0.28 ^a^	0.01 ^a^	−1.95 ^b^
**TPB components**			
**Behavioral beliefs × Evaluation ^A^**	8.10 ^a^	6.34 ^a^	−5.05 ^b^
**Normative beliefs × Motivation to comply ^B^**	−6.96 ^b^	0.65 ^a^	−26.81 ^c^
**Attitude ^C^**	0.64 ^a^	0.00 ^a^	−5.84 ^b^
**Subjective norm ^D^**	−0.41 ^a^	−0.09 ^a^	−2.04 ^b^
**Perceived control ^E^**	0.16 ^a^	−0.65 ^b^	−4.51 ^c^
**Behavioral intention ^E^**	−0.24 ^a^	0.10 ^a^	−4.10 ^b^

Significance for all the comparisons was *p* ≤ 0.001. The 7-point Likert scale were converted from −3 to a +3, with the sole exception of motivational items. * Behavioral beliefs 4/6/7/8 were transformed to positive statements. ^A^ Behavioral beliefs × evaluation ranged between −81 and +81. ^B^ Normative beliefs × motivation to comply ranged between –63 and +63. ^C^ Attitude ranged between −9 and +9. ^D^ Subjective norm ranged between −3 and +3. ^E^ Perceived control and behavioral intention ranged between −6 and +6. ^F^ Behavioral beliefs and normative beliefs ranged between −3 and +3. The lower-case superscript letters indicate the significant differences for each of the clusters found.

## Data Availability

The data presented in this study are available on request from the corresponding author. Although consumer data have been anonymised, data are not publicly available.

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
