# Peer review of "Consumer Attitudes toward Consumption of Meat Products Containing Offal and Offal Extracts"

_foods, 2021, doi:10.3390/foods10071454_

Round 1

Reviewer 1 Report

The manuscript is very good prepared and written. Some comments and suggestions are given below.

Reviewer 2 Report

When assessing the preparation of the work, I assess it very positively. The study was well planned and the results were meticulously presented. Therefore, I recommend this text for publication. Unfortunately, I will not verify the linguistic correctness of the prepared text, so it must be checked by a native speaker anyway.

Reviewer 3 Report

The manuscript “Consumer attitudes toward consumption of meat products containing offal and offal extracts” presents interesting results about consumer perception of “not noble” parts of meat. The interest of the study is highlighted in the context of sustainability and waste reduction, important in the actual context of climate changes”.

The menuscript is well written and presents detail. Only some minor points that I suggest to be addressed before publication:

Introduction – Lines 34-35 - I suggest some modification to this part of the sentence, since the increased consumption in meat is not due to nutritional needs for foods with high contents of animal protein. Maybe the authors want to say that meat consumption continues also due to the nutritional quality (in terms of biological value) of animal protein. But this does not mean that individuals have nutritional needs for high amounts of this protein.

Line 67 - please add a comma between (innate) and cultural

Line 202 - I think this part of the sentence would benefit in re-writen. “perceive its consumption as a barrier” seems to have little meaning. Maybe the authors want to say “perceived as a barrier for consumption the required cleaning step”. Or probably I did not understand what the authors mean with this sentence.

Table 2 - please add the information that small different letters mean significant differences
